# SAMGPT: Text-free Graph Foundation Model for Multi-domain Pre-training and Cross-domain Adaptation

## Abstract

Graphs are able to model interconnected entities in many online services, supporting a wide range of applications on the Web. This raises an important question: How can we train a graph foundational model on multiple source domains and adapt to an unseen target domain? A major obstacle is that graphs from different domains often exhibit divergent characteristics. Some studies leverage large language models to align multiple domains based on textual descriptions associated with the graphs, limiting their applicability to text-attributed graphs. For text-free graphs, very few recent works attempt to align different feature distributions across domains, while generally neglecting structural differences. In this work, we propose a novel Structure Alignment framework for text-free Multi-domain Graph Pre-Training and cross-domain adaptation (SAMGPT). It is designed to learn multi-domain knowledge from graphs originating in multiple source domains, which can then be adapted to address applications in an unseen target domain. Specifically, we introduce a set of *structure tokens* to harmonize structure-based aggregation across source domains during the pre-training phase. Next, for cross-domain adaptation, we design dual prompts, namely, *holistic prompts* and *specific prompts*, which adapt unified multi-domain structural knowledge and fine-grained, domain-specific information, respectively, to a target domain. Finally, we conduct comprehensive experiments on seven public datasets to evaluate and analyze the effectiveness of SAMGPT. (Codes and data are available at https://anonymous.4open.science/r/SAMGPT for anonymous review.)

## CCS Concepts

• **Information systems** → **Web mining**; **Data mining**; • **Computing methodologies** → **Learning latent representations**.

## Keywords

Graph mining, foundation model, multi-domain, pre-training, prompt learning, few-shot learning.

**ACM Reference Format:**
Anonymous Author(s). 2018. SAMGPT: Text-free Graph Foundation Model for Multi-domain Pre-training and Cross-domain Adaptation. In *Proceedings of Make sure to enter the correct conference title from your rights confirmation emai (Conference acronym 'XX).* ACM, New York, NY, USA, 13 pages. https://doi.org/XXXXXXX.XXXXXXX

## 1 Introduction

How to build foundation models has emerged as an important question, paving a plausible path toward artificial general intelligence. In natural language processing, recent works [1, 39] have demonstrated the capabilities of universal foundation models. They are trained on a wide variety of data from multiple domains, and can be further adapted to solve a diverse range of tasks. Other than natural languages, the World Wide Web has become a vast knowledge repository, connecting an enormous amount of entities to form extensive and complex graphs. These graphs enable diverse Web applications, including social network analysis [23, 27], Web mining [2, 52], and recommendation systems [19, 24]. Given the rich graph data on the Web, can we build a universal graph model based on multi-domain graphs, to address various downstream graph-centric applications [15]?

Traditional supervised graph learning struggles to build universal models. These approaches require retraining a new graph neural network (GNN) [7, 12, 40] or graph transformer [29, 54, 62] for each new task, relying on abundant task-specific labeled data. In contrast, more recent graph pre-training methods [10, 26, 41] attempt to learn universal properties from unlabeled graphs in a self-supervised manner, which can be subsequently adapted to a downstream task with some task-specific labels through fine-tuning [11, 26, 41] or prompt learning [17, 37]. However, in most existing graph pre-training approaches, the pre-training and downstream graphs originate from the same dataset [17, 37, 41, 55], a practice we refer to as *single-domain* methods, which fall short of building a universal, *multi-domain* graph model from diverse graph datasets.

**Research problem.** Thus, it is crucial to pre-train a graph model on a wide range of multi-domain (i.e., multi-dataset) graphs and achieve cross-domain adaptation. However, graph structures from different datasets often exhibit markedly distinct characteristics. For instance, the structural patterns in a social network might not be directly applicable to a citation or e-commerce graph. Such diversity poses significant challenges in integrating graphs from multiple domains and adapting prior knowledge to different domains. Although some studies have explored cross-domain adaptation from a single source domain [4, 8, 42, 44, 53], they do not exploit multiple source domains. Another line of work [14, 38, 50] employs large language models to extract and utilize multi-domain knowledge based on textual descriptions associated with the graphs, using text as an universal medium to bridge different domains. However, this limits their applicability to text-attributed graphs [46, 66] and cannot be extended to general graphs without textual descriptions. Few recent studies [61, 65] have explored multi-domain pre-training on text-free graphs, but they focus on aligning the divergent feature spaces and homophily patterns across multi-domain graphs, while overlooking the structural differences across domains.

**Challenges and insights.** In this paper, we propose SAMGPT, a graph foundation model with **S**tructural **A**lignment for text-free

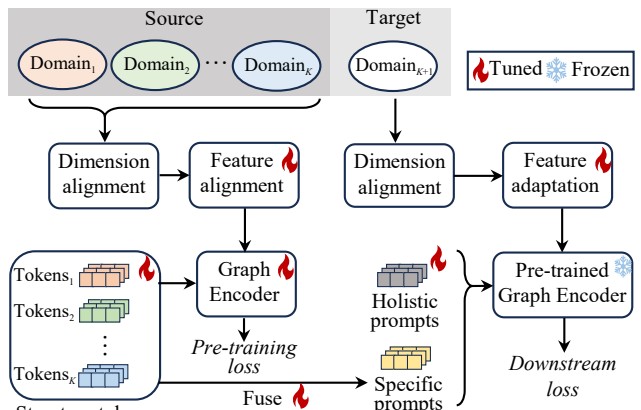

Figure 1: Motivation of SAMGPT.

**M**ulti-domain **G**raph **P**re-**T**raining, to facilitate cross-domain adaptation. This is non-trivial due to two key challenges.

First, *how do we harmonize structural variance across multiple domains during pre-training?* Graphs from different domains often exhibit distinct structural and topological characteristics, as depicted in Appendix C. Consequently, merging multi-domain graphs without proper structure alignment during pre-training can lead to interference rather than synergistic benefits, resulting in suboptimal performance. In SAMGPT, we propose the notion of *structure tokens* to align structural distributions across multiple domains, as shown in Fig. 1(a). Specifically, each domain is equipped with a series of structure tokens, which modify the structure-based aggregation in each layer of the graph encoder. These tokens are learnable vectors that capture domain-specific structural patterns, enabling the model to accommodate the unique structural characteristics of each domain during pre-training.

Second, *how do we adapt multi-domain structural prior knowledge to cross-domain downstream tasks?* Multi-domain prior knowledge includes not only holistic knowledge across source domains, but also domain-specific knowledge from each domain. Therefore, in SAMGPT, we propose dual *structural prompts*, comprising a set of *holistic prompts* and a set of *specific prompts*, thus facilitating the adaptation of both holistic and domain-specific knowledge to downstream tasks, as illustrated in Fig. 1(b). On one hand, the holistic prompts consist of learnable vectors that holistically align the target domain's structural characteristics with the unified pre-trained knowledge from all source domains. On the other hand, specific prompts integrate multi-domain structure tokens in a learnable mixture to align the target domain with knowledge from each source domain, capturing domain-specific structural information for finer-grained adaptation.

**Contributions.** In summary, we make the following contributions in this work. (1) We propose SAMGPT, a text-free graph foundation model with structure alignment for multi-domain graph pre-training and cross-domain adaptation. (2) For pre-training, we propose structure tokens to align structural distributions across domains, training a universal foundation model with multi-domain graphs. (3) For downstream adaptation, we propose a dual-prompt

strategy, using holistic prompts to leverage holistic prior structural knowledge and specific prompts to facilitate finer-grained, domain-specific structural adaptation. (4) We conduct extensive experiments on seven benchmark datasets. The results demonstrate that SAMGPT achieves superior performance compared to state-of-the-art methods.

## 2  Related Work

We review related literature on pre-training, cross-domain transfer learning, and multi-domain pre-training for graph data.

**Graph pre-training.** Graph pre-training methods aim to extract inherent properties of graphs, often utilizing self-supervised learning approaches, which can be either generative [9, 10, 13] or contrastive [41, 49, 51]. The pre-trained model is then employed to address downstream tasks through fine-tuning [26, 41, 55] or parameter-efficient adaptation methods, notably prompt-based learning [5, 17, 36, 57]. However, these methods typically assume that the pre-training and downstream graphs originate from the same domain, such as different subgraphs of a large graph [55, 58] or collections of similar graphs within the same dataset [10, 26], failing to account for multiple domains in either pre-training or downstream graphs.

**Graph cross-domain transfer.** This line of work aims to transfer single-source domain knowledge to a different target domain by leveraging domain-invariant properties across domains [4, 8, 42, 44]. However, they rely exclusively on a single source domain, failing to harness the extensive knowledge available across multiple domains. Additionally, these approaches are often tailored to specific tasks or domains [4, 8, 42, 44], limiting their generalization.

**Multi-domain graph pre-training.** In the context of multi-domain pre-training and cross-domain adaptation, recent works [14, 38, 50] utilize large language models to align node features from different domains through textual descriptions, thereby limiting their applicability to text-attributed graphs [47, 63, 66]. For graphs without textual attributes, GraphControl [68] applies ControlNet [64] to incorporate target domain node features with the pre-trained model, while neglecting the alignment among multiple source domains. Another recent study proposes GCOPE [65], which employ domain-specific virtual nodes connected to nodes within each domain, facilitating the alignment of feature distribution and homophily patterns. Meanwhile, MDGPT [61] pre-trains domain-specific tokens to align feature semantics across various domains. However, these studies do not account for structural variance across different domains, hindering their effectiveness in integrating multi-domain knowledge. On a related front, multi-task pre-training techniques [45, 60] employ pretext tokens for each pre-training task. It is important to note that they address a distinct problem, aiming to overcome potential interference among multiple tasks within a single domain, rather than interference across multiple domains.

## 3  Preliminaries

In this section, we provide technical background, and outline the scope of our work.

**Graph encoder.** A *graph* is defined as $G = (V, E, \mathbf{X})$, where $V$ is the set of nodes, $E$ is the set of edges, and $\mathbf{X} \in \mathbb{R}^{|V| \times d}$ is the node

feature matrix with each row $\mathbf{x}_i$ representing the feature vector of node $v_i \in V$. A collection of graphs is denoted as $\mathcal{G}$.

Message-passing GNNs are a common choice for encoding graph representations [48]. Specifically, each node updates its embedding by receiving and aggregating features or embeddings from its neighbors. By stacking such message-passing layers, information can propagate recursively throughout the graph. Therefore, the node embeddings are encoded based on both input features and graph structure. Let us denote the embedding of node $v$ at the $l$-th layer as $\mathbf{h}_v^l$, which is derived from the features or embeddings in the preceding layer as follows.

$$\mathbf{h}_v^l = \mathrm{Aggr}(\mathbf{h}_v^{l-1}, \{\mathbf{h}_u^{l-1} : u \in \mathcal{N}_v\}; \theta^l), \qquad (1)$$

where $\mathcal{N}_v$ denotes the set of neighboring nodes of $v$, $\theta^l$ represents the learnable parameters in layer $l$, and $\mathrm{Aggr}(\cdot)$ stands for the neighborhood aggregation function. In the first layer, the node embedding $\mathbf{h}_v^0$ is initialized as the input feature vector $\mathbf{x}_v$. We denote the output node embedding after the last layer as $\mathbf{h}_v$, which is a row in the node embedding matrix $\mathbf{H}$. Overall, the multi-layer message-passing process can be abstracted as a *graph encoder*, as follows.

$$\mathbf{H} = \mathrm{GE}(G, \mathbf{X}; \Theta), \qquad (2)$$

where $\mathrm{GE}$ denotes a graph encoder, $\Theta = \{\theta^1, \theta^2, \ldots\}$ is the full set of trainable parameters for the graph encoder.

**Multi-domain pre-training with feature alignment.** Consider a set of unlabeled graphs $\mathcal{G}_S = \{G_1, G_2, \ldots, G_K\}$ for pre-training, where each graph $G_i$ belongs to a specific *source domain* $D_{S_i} \in \mathcal{D}_S$. Thus, we have graph-domain pairs $\{(G_i, D_{S_i}) : i \in \{1, 2, \ldots, K\}\}$.

As different domains exhibit distinct feature distributions, previous works [56, 65] have proposed solutions to align feature dimensions and semantics, which can be directly employed in our work. Given a graph $G_i = (V_i, E_i, \mathbf{X}_i)$ from the source domain $D_{S_i}$, we first align the dimensions of its feature matrix:

$$\tilde{\mathbf{X}}_i = \mathrm{DAL}_{S_i}(\mathbf{X}_i), \qquad (3)$$

where $\mathrm{DAL}_{S_i} : \mathbb{R}^{|V| \times d_{S_i}} \to \mathbb{R}^{|V| \times \tilde{d}}$ is the dimension alignment function for domain $D_{S_i}$, transforming the original dimension $d_{S_i}$ to a common dimension $\tilde{d}$ across domains. We implement $\mathrm{DAL}$ as singular value decomposition [34] following prior art [56, 65]. Next, given the source-domain graphs $\mathcal{G}_S$ with their dimension-aligned features $\tilde{\mathcal{X}}_S = \{\tilde{\mathbf{X}}_i : G_i \in \mathcal{G}_S\}$, we further align the features to unify their semantic space across various domains. Letting $\mathrm{FAL}$ denote the feature alignment procedure, we pre-train a graph encoder with feature alignment:

$$\mathbf{H}^{\mathrm{FAL}} = \mathrm{GE}(\mathrm{FAL}(\mathcal{G}_S, \tilde{\mathcal{X}}_S; \Psi); \Theta), \qquad (4)$$

where $\Psi$ denotes learnable parameters in $\mathrm{FAL}$, and $\mathbf{H}^{\mathrm{FAL}}$ is the output node embedding matrix with feature alignment. While any feature alignment model can be employed [56, 65], we follow the work of Yu et al. [56] due to its superior performance.

**Cross-domain task with feature adaptation.** For each downstream task, consider a set of graphs $\mathcal{G}_T$ belonging to a target domain $D_T$. The task is *cross-domain* if the target domain is *unseen* during pre-training, *i.e.*, $\forall i \; D_T \neq D_{S_i}$. Again, since the target domain may exhibit different feature characteristics from the source domains, previous works [61, 65] have proposed feature adaptation

strategies to transfer prior multi-domain knowledge to the target domain, which can be directly integrated into our work. Specifically, we first employ the same dimension alignment method used in the pre-training phase, transforming the feature matrix of a downstream graph $G = (V, E, \mathbf{X}) \in \mathcal{G}_T$ to $\tilde{\mathbf{X}} = \mathrm{DAL}_T(\mathbf{X})$. We then employ a feature adaptation technique $\mathrm{FAD}$ to adapt the pre-trained model to the target domain, as follow.

$$\mathbf{H}^{\mathrm{FAD}} = \mathrm{GE}(\mathrm{FAD}(\mathcal{G}, \tilde{\mathbf{X}}; \Gamma); \Theta_{\mathrm{pre}}), \qquad (5)$$

where $\Gamma$ denotes the learnable parameters in $\mathrm{FAD}$, and $\Theta_{\mathrm{pre}}$ is the pre-trained weights in graph encoder $\mathrm{GE}$. Here we implement $\mathrm{FAD}$ following Yu et al. [56], which is paired with the feature alignment method in pre-training.

**Our scope: Few-shot classification.** For the downstream applications, we aim to solve *few-shot* node and graph classification tasks. For node classification, given a graph $G = (V, E, \mathbf{X}) \in \mathcal{G}_T$, each node $v \in V$ is associated with a label $y \in Y$, where $Y$ denotes the set of node classes. For graph classification over a set of graphs $\mathcal{G}_T$, each graph $G \in \mathcal{G}_T$ is associated with a label $y \in Y$, where $Y$ denotes the set of graph classes. An $m$-shot classification task consists of only $m$ labeled examples per class, along with an arbitrary number of unlabeled examples for testing.

In particular, we focus on *low-shot* settings, where $m$ is a small number (*e.g.*, $m \leq 5$), reflecting real-world applications where labeled data are expensive or difficult to obtain. Due to the parameter-efficient nature of prompt learning, many previous methods for prompt learning on graphs [17, 37, 57, 61, 65] also emphasize this setting. It is expected that, as more task-specific labeled data become available, conventional fine-tuning or supervised approaches may become sufficient.

## 4 Proposed Approach: SAMGPT

In this section, we present SAMGPT, beginning with an overview and then delving into the details of multi-domain pre-training and cross-domain adaptation.

### 4.1 Overall Framework

SAMGPT consists of two phases: multi-domain pre-training, and cross-domain adaptation, as shown in Fig. 2.

In the pre-training phase, as depicted in Fig. 2(a), we first align the feature distributions from multiple source domains following previous work [61, 65]. Next, we introduce a set of *structure tokens* designed to align the structural distributions across diverse domains. These tokens are domain-specific and are integrated into each layer of the graph encoder, modifying the structure-based aggregation at each layer. Finally, the structure token-enhanced graph encoder is pre-trained using a self-supervised loss based on a universal task template [17].

In the adaptation phase, as shown in Fig. 2(b), we first align the feature dimension of the target domain with that of the source domains. Then, we introduce *dual prompts*. The first type, *holistic prompts*, are learnable vectors that integrate the target domain with the holistic structural knowledge from all source domains. The second type, *specific prompts*, comprise learnable mixtures of pre-trained structure tokens that incorporate domain-specific topological information tailored to the target domain. These prompts are

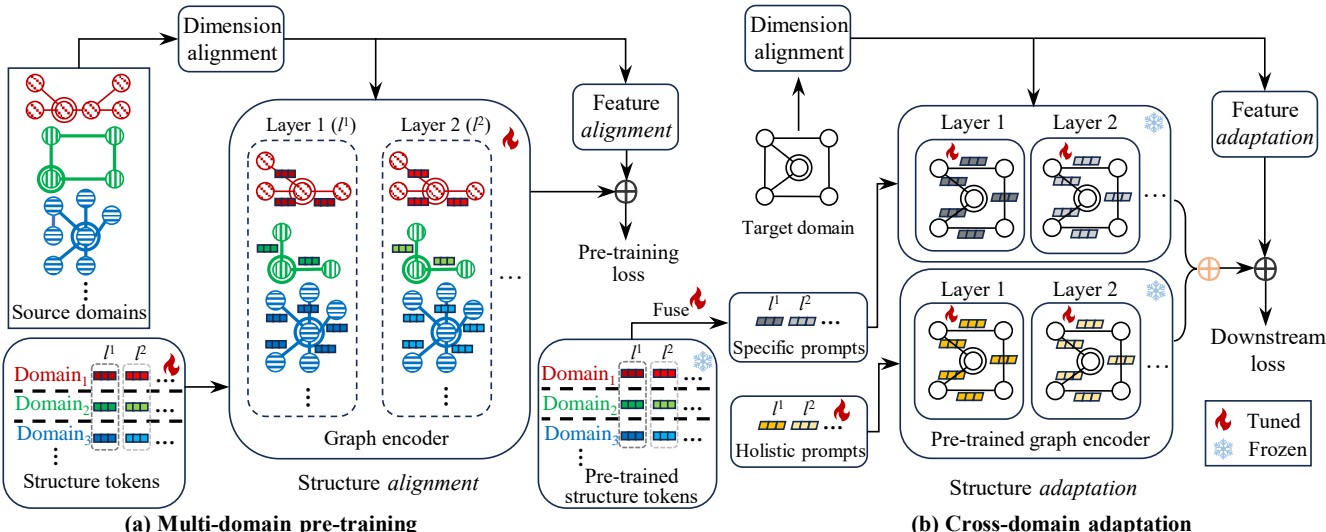

**Figure 2: Overall framework of SAMGPT.**

applied to each layer of the graph encoder to adjust the structure-based aggregation, while keeping the pre-trained weights of the graph encoder frozen.

## 4.2 Multi-domain Graph Pre-training with Structure Alignment

As defined in Sect. 3, we are given a set of pre-training graphs from multiple source domains, $\mathcal{G}_S$. As both the features and structures of these domains can exhibit divergent distributions, effective integration of these multi-domain graphs requires aligning both. As our work focuses on structure alignment, we follow previous feature alignment methods [56, 65], as outlined in the preliminaries.

**Structure alignment.** Recall that in the graph encoder, node representations are updated layer-wise through a structure-based aggregation. Each layer captures different levels of structural information. For example, the first layer aggregates one-hop neighborhood information, while the second layer incorporates a broader two-hop neighborhoods. These layer-wise structural patterns may vary significantly across domains.

Therefore, to unify the structural characteristics in multiple source domains, we introduce learnable *structure tokens*. For each domain $D_{S_i}$, we inject a series of structure tokens $\mathcal{T}_{S_i} = \{\mathbf{t}_{S_i}^l : l \in \{1, \ldots, L\}\}$ into the graph encoder, where $L$ denotes the number of layers. Specifically, when encoding the graph $G_i = (V_i, E_i, \tilde{\mathbf{X}}_i)$ in $D_{S_i}$, we assign structure token $\mathbf{t}_{S_i}^l$ to the $l$-th layer, guiding structure-based aggregation:

$$\mathbf{h}_v^l = \text{Aggr}(\mathbf{h}_v^{l-1}, \{\mathbf{t}_{S_i}^l \odot \mathbf{h}_u^{l-1} : u \in \mathcal{N}_v\}; \theta^l), \ \forall v \in V_i, \quad (6)$$

where $\odot$ represents element-wise multiplication. Note that the graph encoders for feature alignment and structure alignment on all graphs share the same parameters $\Theta$. Let $\mathbf{H}_i^{\text{SAL}}$ denote the structure-aligned output node embedding matrix for $G_i$ in $D_{S_i}$, following the aggregation in Eq. (6). In general, each source domain is attached with its own set of structure tokens, which are applied to modify

the aggregation on the graph in the corresponding domain. By stacking the structure-aligned output matrix across graphs in all domains, we obtain the overall structure-aligned embedding matrix, $\mathbf{H}^{\text{SAL}} = \text{Stack}(\mathbf{H}_1^{\text{SAL}}, \ldots, \mathbf{H}_K^{\text{SAL}})$.

Finally, we fuse $\mathbf{H}^{\text{SAL}}$ with $\mathbf{H}^{\text{FAL}}$ in Eq. (4) to obtain the multi-domain node embedding matrix $\mathbf{H}$, incorporating both feature and structure alignment, as shown below.

$$\mathbf{H}^{\text{AL}} = \mathbf{H}^{\text{FAL}} + \alpha \mathbf{H}^{\text{SAL}}, \quad (7)$$

where $\alpha > 0$ is a hyperparameter.

**Pre-training loss.** We leverage a universal task template based on subgraph similarity calculation [17, 57], which ensures compatibility across different tasks such as node classification and graph classification. As demonstrated in GraphPrompt+ [57], prevailing contrastive pre-training objectives can be unified under this template, making them suitable choices for the pre-training loss in SAMGPT. In general, we can adopt the following form of contrastive loss in pre-training.

$$\mathcal{L}_{\text{pre}}(O; \Theta, \mathcal{T}, \Psi) = -\sum_{o \in O} \ln \frac{\sum_{a \in \text{Pos}_o} \exp(\text{sim}(\mathbf{h}_a, \mathbf{h}_o)/\tau)}{\sum_{b \in \text{Neg}_o} \exp(\text{sim}(\mathbf{h}_b, \mathbf{h}_o)/\tau)}, \quad (8)$$

where $O$ denotes the set of observed graph element in pre-training, $a \in \text{pos}_o, b \in \text{neg}_o$ represent the positive or negative instance of $o$, respectively, and $\mathbf{h}_o, \mathbf{h}_a, \mathbf{h}_b$ are their corresponding embeddings. Furthermore, $\text{sim}(\cdot, \cdot)$ is a similarity function, such as cosine similarity [28] in our implementation, and $\tau > 0$ is a temperature hyperparameter. Note that SAMGPT is flexible in the materialization of $o, a, b$ to realize different contrastive losses [57]. Our experiments adopt GraphCL [55], where $a$ is the original graph $G$, and $o, b$ represent two different augmentations of $G$. Hence, $\mathbf{h}_o, \mathbf{h}_a, \mathbf{h}_b$ are the corresponding graph embeddings, which can be obtained through a readout operation [17] on the aligned node embeddings in $\mathbf{H}^{\text{AL}}$.

The pre-training loss is optimized by updating the weights of graph encoder $\Theta$, structure tokens across all source domains $\mathcal{T} = \{\mathcal{T}_{S_1}, \ldots, \mathcal{T}_{S_K}\}$, and feature alignment parameters $\Psi$.

## 4.3 Cross-domain Structure Adaptation

Beyond multi-domain pre-training, another challenge lies in cross-domain adaptation. Given a model pre-trained on graphs $\mathcal{G}_S$ from source domains $\mathcal{D}_S$, we aim to adapt it to a downstream task on graphs $\mathcal{G}_T$ from a target domain $D_T \notin \mathcal{D}_S$. As this work focuses on structure adaptation, we directly apply previous work [56] for feature adaptation, as outlined in Sect. 3.

For structure adaptation, we propose *dual prompts*, consisting of *holistic prompts* and *specific prompts*. On one hand, the holistic prompts are designed to holistically utilize the pre-trained structural knowledge from all source domains. On the other hand, the specific prompts combine multi-domain structure tokens through a learnable mixture, adapting fine-grained, domain-specific structural knowledge to the target domain.

**Holistic prompts.** To transfer the holistic multi-domain structural knowledge to a downstream task, we propose a set of holistic prompts designed to align the target domain $D_T$ with the model pre-trained on the source domains $\mathcal{D}_S$. Like any pre-training framework, we encode a downstream graph $G = (V, E, \tilde{\mathbf{X}})$ using the pre-trained graph encoder with frozen layer-wise weights $\Theta_{\text{pre}} = \{\theta_{\text{pre}}^1, \ldots, \theta_{\text{pre}}^L\}$. However, the key difference is we inject a series of learnable vectors $\mathcal{P}_{\text{hol}} = \{\mathbf{p}_{\text{hol}}^1, \ldots, \mathbf{p}_{\text{hol}}^L\}$ as holistic prompts into the downstream structure-based aggregation:

$$\mathbf{h}_v^l = \text{Aggr}(\mathbf{h}_v^{l-1}, \{\mathbf{p}_{\text{hol}}^l \odot \mathbf{h}_u^{l-1} : u \in \mathcal{N}_v\}; \theta_{\text{pre}}^l), \ \forall v \in V. \quad (9)$$

The final layer outputs a holistic node embedding matrix for the downstream graph $G$, denoted as $\mathbf{H}^{\text{hol}}$.

**Specific prompts.** In contrast to the holistic prompts, specific prompts are designed to adapt structural knowledge specific to each source domain. Since knowledge from related source domains is likely to be more applicable, it is essential to align the target domain with different source domains to varying extents, prioritizing the most relevant ones. Consequently, we define specific prompts as $\mathcal{P}_{\text{spe}} = \{\mathbf{p}_{\text{spe}}^1, \ldots, \mathbf{p}_{\text{spe}}^L\}$, which will also be injected into different layers of the pre-trained graph encoder. Specifically, in the $l$-th layer, $\mathbf{p}_{\text{spe}}^l$ is a combination of $\{\mathbf{t}_{S_1}^l, \ldots, \mathbf{t}_{S_K}^l\}$, the pre-trained structure tokens in the corresponding layer across all source domains $D_{S_i} \in \mathcal{D}_S$. Formally, we define

$$\mathbf{p}_{\text{spe}}^l = \sum_{i=1}^K \lambda_i^l \mathbf{t}_{S_i}^l, \quad (10)$$

where $\Lambda^l = \{\lambda_1^l, \ldots, \lambda_K^l\}$ are learnable coefficients. Thus, the full set of learnable parameters for the specific prompts is $\Lambda = \{\Lambda^1, \ldots, \Lambda^L\}$. Subsequently, specific prompts modify the structure-based aggregation in the same way as in Eq. (9), while freezing the pre-trained weights of the graph encoder. Similarly, we denote the output node embedding matrix based on the specific prompts as $\mathbf{H}^{\text{spe}}$.

**Prompt tuning.** To leverage both holistic multi-domain and domain-specific structural knowledge from the pre-trained model, we fuse the output embedding matrices obtained via holistic prompts and specific prompts as follows.

$$\mathbf{H}^{\text{SAD}} = \mathbf{H}^{\text{hol}} + \beta \mathbf{H}^{\text{spe}}, \quad (11)$$

where $\beta > 0$ is a hyperparameter. Further incorporating feature adaptation in Eq. (5), we obtain the overall node embedding matrix

with both feature and structure adaptations, given by

$$\mathbf{H}^{\text{AD}} = \mathbf{H}^{\text{FAD}} + \alpha \mathbf{H}^{\text{SAD}}. \quad (12)$$

Here, $\alpha$ is the same hyperparameter used in Eq. (7), as both share the objective of integrating the feature and structure counterparts.

For downstream node and graph classification tasks, the loss function $\mathcal{L}_{\text{down}}$ is formulated based on the same task template centered on subgraph similarity [17], akin to the pre-training loss $\mathcal{L}_{\text{pre}}$. Let $\Omega = \{(x_1, y_1), (x_2, y_2), \ldots\}$ represent the labeled training set, where each $x_i$ is either a node or graph instance, and $y_i \in Y$ is its respective class from the set $Y$. Subsequently, we optimize the following cross-domain adaptation loss:

$$\mathcal{L}_{\text{down}}(\Omega; \mathcal{P}_{\text{hol}}, \Lambda, \Gamma) = -\sum_{(x_i, y_i) \in \Omega} \ln \frac{\exp(\text{sim}(\mathbf{h}_{x_i}, \mathbf{h}_{y_i})/\tau)}{\sum_{y \in Y} \exp(\text{sim}(\mathbf{h}_{x_i}, \mathbf{h}_y)/\tau)}. \quad (13)$$

Here, $\mathbf{h}_{x_i}$ represents the adapted embedding of the node or graph $x_i$ based on $\mathbf{H}^{\text{AD}}$, where a readout operation on $\mathbf{H}^{\text{AD}}$ is required if $x_i$ is a graph. Additionally, $\mathbf{h}_y$ denotes the prototype embedding for class $y$, which is calculated as the average embeddings of all training instances of class $y$.

We outline the key steps for prompt tuning in Algorithm 1, Appendix A and assess its complexity in Appendix B.

## 5 Experiments

In this section, we conduct experiments to assess the performance of SAMGPT and analyze its empirical results.

### 5.1 Experimental Setup

**Datasets.** We conduct experiments on seven benchmark datasets. (1) *Cora* [22], (2) *Citeseer* [32] and (3) *Pubmed* [32] are scientific paper citation networks from different fields, such as computer science and biomedical research. Nodes represent academic publications and edges denote citation relationships. (4) *Photo* [33] and (5) *Computers* [21] are both e-commerce networks from Amazon in different categories, namely, photography and computer related products. Nodes represent products and edges signify frequent co-purchases between products. (6) *Facebook* [30] is a Web graph, where nodes represent official Facebook pages while the links are mutual likes between these pages. (7) *LastFM* [31] is a social network, where nodes denote users and edges represent interactions such as follower relationships. Note that each domain comprises a single graph. We present additional details of these datasets in Appendix C.

**Setup of pre-training and downstream tasks.** Following previous work [61, 65], we treat each dataset as a distinct domain. Among the seven datasets (or domains), we use each of them as the target domain while using the remaining six as source domains. We conduct $m$-shot *node classification* and *graph classification*, where $m$ labeled nodes or graphs per class are randomly selected for downstream prompt tuning. Given that each dataset comprises a single graph, performing graph classification on whole graphs is not feasible. Therefore, following previous works [18, 58, 59], we generate a series of graphs by constructing ego-networks centered on the labeled nodes within each dataset, and set up graph classification on these ego-networks, with each network labeled according to its central node. Note that the graph encoder is pre-trained only once

**Table 1: Accuracy (%) of one-shot *node classification* with standard deviations. Each column represents a target domain, using other columns as source domains. The best method in each column is bolded, and the runner-up is underlined.**

| Method\Target domain | Cora | Citeseer | Pubmed | Photo | Computers | Facebook | LastFM |
|---|---|---|---|---|---|---|---|
| GCN | 29.53 ± 7.56 | 26.29 ± 6.50 | 23.32 ± 11.56 | 26.96 ± 12.94 | 24.40 ± 5.62 | 20.45 ± 5.62 | 9.21 ± 3.11 |
| GAT | 24.27 ± 9.26 | 21.56 ± 8.09 | 22.28 ± 9.76 | 17.85 ± 10.22 | 23.03 ± 12.12 | 29.27 ± 6.47 | 9.01 ± 2.61 |
| DGI | 33.40 ± 10.48 | 25.80 ± 8.27 | 47.22 ± 9.50 | 30.89 ± 10.54 | 25.75 ± 12.45 | 34.36 ± 9.57 | 14.14 ± 6.31 |
| GraphCL | 27.72 ± 9.37 | 35.02 ± 8.46 | 48.89 ± 9.03 | 34.78 ± 11.56 | 23.79 ± 12.28 | 34.85 ± 7.07 | 18.93 ± 7.32 |
| GPPT | 27.18 ± 4.88 | 25.90 ± 4.68 | 39.82 ± 8.79 | 31.58 ± 10.27 | 19.94 ± 9.61 | 34.73 ± 3.99 | 20.98 ± 3.98 |
| GraphPrompt | 28.26 ± 12.68 | 32.51 ± 8.73 | 47.47 ± 9.15 | 48.11 ± 9.89 | 42.82 ± 11.67 | 40.44 ± 9.68 | 19.84 ± 7.23 |
| GPF | 32.17 ± 6.56 | 36.79 ± 7.70 | 41.28 ± 8.14 | 47.47 ± 8.19 | 35.75 ± 7.12 | 40.45 ± 6.34 | 27.26 ± 5.50 |
| Hassani | 33.35 ± 6.93 | 33.66 ± 7.24 | 39.87 ± 8.16 | 48.48 ± 7.07 | 39.99 ± 7.91 | 37.70 ± 5.79 | 27.16 ± 4.94 |
| GCOPE | 35.62 ± 11.93 | **38.33** ± 9.28 | 45.38 ± 9.87 | 52.87 ± 9.19 | 45.65 ± 10.69 | 40.63 ± 8.50 | 28.84 ± 7.59 |
| SAMGPT | **47.80** ± 11.88 | 36.38 ± 9.10 | **50.25** ± 10.43 | **58.71** ± 8.69 | **48.22** ± 8.17 | **42.70** ± 8.73 | **33.36** ± 8.11 |

**Table 2: Accuracy (%) of one-shot *graph classification* with standard deviations. Each column represents a target domain, using other columns as source domains. The best method in each column is bolded, and the runner-up is underlined.**

| Method\Target domain | Cora | Citeseer | Pubmed | Photo | Computers | Facebook | LastFM |
|---|---|---|---|---|---|---|---|
| GCN | 30.64 ± 10.31 | 26.90 ± 7.15 | 38.84 ± 11.82 | 15.60 ± 8.77 | 21.94 ± 14.51 | 31.33 ± 9.47 | 28.83 ± 9.60 |
| GAT | 27.80 ± 7.85 | 27.50 ± 7.13 | 21.66 ± 8.70 | 15.74 ± 7.62 | 16.02 ± 13.46 | 21.20 ± 7.31 | 27.80 ± 7.85 |
| InfoGraph | 34.98 ± 10.15 | 35.87 ± 9.84 | 48.67 ± 12.29 | 25.70 ± 11.73 | 19.02 ± 14.09 | 31.26 ± 9.65 | 23.29 ± 7.99 |
| GraphCL | 42.70 ± 10.64 | 36.66 ± 8.67 | 47.53 ± 11.52 | 33.07 ± 12.31 | 16.02 ± 13.47 | 21.99 ± 13.00 | 21.30 ± 10.45 |
| GraphPrompt | 37.38 ± 14.03 | 36.66 ± 9.19 | **49.55** ± 10.25 | 50.79 ± 12.31 | 43.09 ± 11.45 | 41.71 ± 10.61 | 32.62 ± 8.54 |
| GPF | 39.62 ± 8.52 | 36.73 ± 7.66 | 45.08 ± 10.36 | 47.57 ± 10.16 | 35.70 ± 8.71 | 34.84 ± 5.14 | 34.31 ± 7.05 |
| Hassani | 36.86 ± 10.74 | 35.78 ± 8.80 | 43.97 ± 13.27 | 41.55 ± 13.08 | 29.49 ± 13.86 | 35.57 ± 9.00 | 25.39 ± 8.14 |
| GCOPE | 38.85 ± 10.99 | **39.93** ± 9.82 | 47.05 ± 11.74 | 53.93 ± 9.74 | 45.60 ± 10.96 | 40.26 ± 9.53 | 34.68 ± 7.70 |
| SAMGPT | **55.35** ± 13.62 | 38.75 ± 9.40 | 48.69 ± 10.16 | **58.75** ± 11.67 | **48.72** ± 11.18 | **43.71** ± 9.54 | **48.28** ± 9.72 |

for each set of source domains, and subsequently utilized across all downstream tasks. We generate 100 $m$-shot tasks for both node classification and graph classification by repeatedly sampling $m$ labeled nodes/graphs per class for 100 times. Each task is executed with five different random seeds, leading to a total of 500 outcomes for each classification type. We use accuracy as the evaluation metric, as each task is class-balanced [16, 17, 43, 57], and report the average accuracy and standard deviation over these 500 outcomes.

**Baselines.** We compare the performance of SAMGPT against state-of-the-art methods in four broad groups, as follow. (1) *End-to-end graph neural networks*: GCN [12] and GAT [40] aggregate information from neighboring nodes to update node representations. For each task, they are trained from scratch in a supervised fashion without pre-training. (2) *Graph pre-training models*: DGI [41], Info-Graph [35][1] and GraphCL [55] first pre-train a graph encoder to capture the inherent properties of the graphs, and then fine-tune a classifier on the downstream task while freezing the pre-trained model. GPPT[2] [36], GPF[5] and GraphPrompt [17] employ a universal task template to unify self-supervised pre-training and downstream tasks, and tune a single prompt on downstream tasks. (3) *Graph cross-domain model*: Hassani [8] pre-trains a GNN on a single

source domain by incorporating both contextual and topological views, facilitating cross-domain adaptation for downstream tasks. (4) *Multi-domain pre-training model*: GCOPE [65] performs multi-domain pre-training via self-supervised learning, and subsequently adapts to cross-domain tasks through either fine-tuning a classification head or prompt tuning. We opt for fine-tuning as it yields superior performance.

Note that the above graph pre-training and cross-domain approaches are originally designed for pre-training on a single source domain. For a fair comparison, we directly merge the multi-domain graphs and apply dimension alignment for them, as in SAMGPT. Further descriptions of the baselines are provided in Appendix D, along with implementation and configuration details for baselines and SAMGPT in Appendix E.

## 5.2 Few-shot performance evaluation

We first compare SAMGPT and baseline methods on one-shot node and graph classification tasks, and then explore the effect of increasing the number of shots.

**One-shot performance.** Tables 1 and 2 show the results of one-shot node and graph classification tasks. Our observations are as follows. First, SAMGPT achieves outstanding performance in both node and graph classification across various target domains, demonstrating the effectiveness of our proposed structure tokens in multi-domain pre-training and dual prompts in cross-domain adaptation.

---

[1]Original DGI only operates at the node level, while InfoGraph extends it to the graph level. We apply DGI to node classification, and InfoGraph to graph classification.
[2]GPPT is tailored for node classification task and is not applicable to graph classification. Thus, in our experiments, we only use GPPT for node classification.

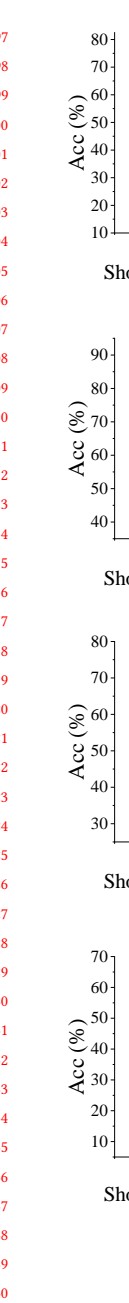

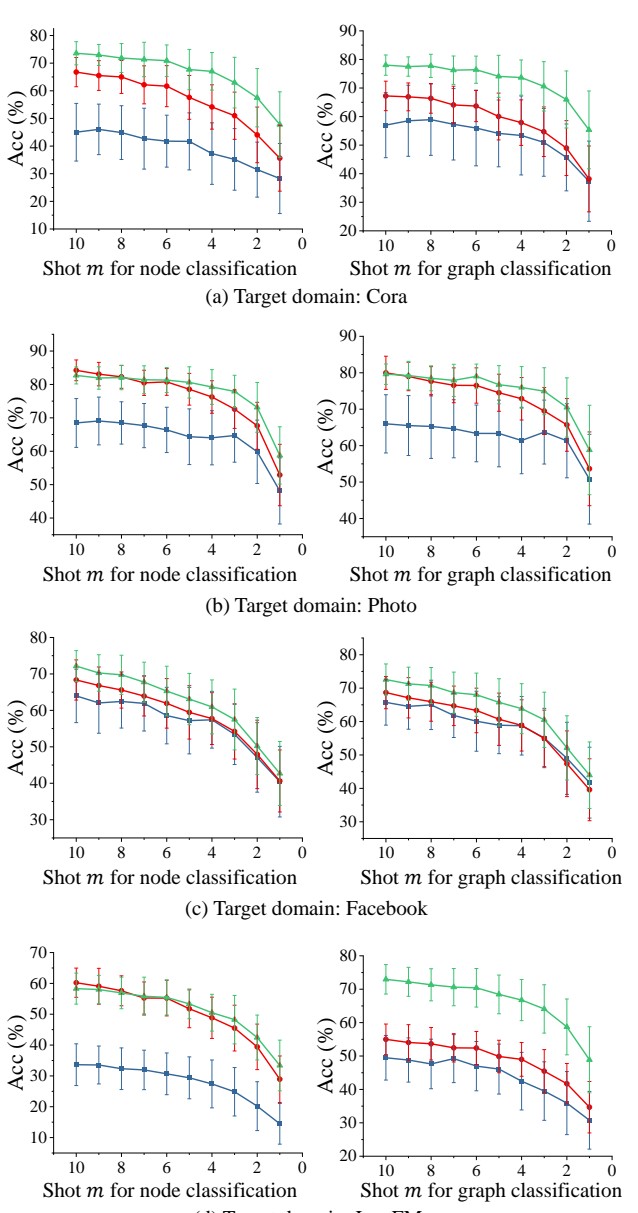

(a) Target domain: Cora

(b) Target domain: Photo

(c) Target domain: Facebook

(d) Target domain: LastFM

■ GraphPrompt    ● GCOPE    ▲ SAMGPT

**Figure 3: Impact of number of shots on node and graph classification on four target domains.**

Refer to the ablation studies in Sect. 5.3 for the quantitative contributions of these components. Second, another text-free multi-domain pre-training method, GCOPE, significantly lags behind SAMGPT because it only performs alignment and adaptation on feature and homophily patterns, without accounting for structural differences across domains. This further emphasizes the importance of our structure tokens and dual prompts. Third, graph pre-training methods generally outperform the end-to-end GCN and GAT, showcasing the benefits of pre-training on unlabeled graphs.

**Table 3: Data ablation study with an increasing number of source domains.**

| Method | Number of source domains | | | |
|---|---|---|---|---|
| | 1 | 2 | 3 | 4 |
| GraphPrompt | $35.53_{\pm12.06}$ | $37.13_{\pm11.79}$ | $36.90_{\pm11.23}$ | $38.54_{\pm11.84}$ |
| GCOPE | $39.47_{\pm12.14}$ | $36.63_{\pm 9.46}$ | $35.28_{\pm11.99}$ | $38.61_{\pm12.74}$ |
| SAMGPT | $\mathbf{40.43}_{\pm11.00}$ | $\mathbf{41.97}_{\pm11.01}$ | $\mathbf{42.30}_{\pm11.56}$ | $\mathbf{45.95}_{\pm12.96}$ |

**Few-shot performance.** To evaluate the performance of SAMGPT with more labeled data, we vary the number of shots, $m$, in both node and graph classification tasks. We compare SAMGPT to two competitive baselines, GraphPrompt and GCOPE, with results reported in Fig. 3, where error bars represent the standard deviation. We observe that SAMGPT consistently outperforms the baselines in low-shot settings (*e.g.*, $m \leq 5$). When further increasing the number of shots, SAMGPT still performs best in general, although it may be on par with GCOPE in some cases when $m$ approaches 10. This is not surprising, since the advantage of models may diminish as more supervision becomes available.

## 5.3 Ablation Studies

To gain deeper insight into the impact of each component in SAMGPT, we conduct two ablation studies.

**Data ablation.** We evaluate the impact of incorporating more source domains by incrementally adding *Citeseer*, *LastFM*, *Photo*, and *Facebook*, in this order, to the pre-training, while fixing *Cora* as the target domain. We present 1-shot node classification results of SAMGPT and two competitive baselines, namely, GraphPrompt and GCOPE, in Table 3. On the $x$-axis, 1 represents using *Citeseer* as the single source domain, while 2 represents using *Citeseer* and *LastFM* as source domains, etc.

We make the following observations. First, SAMGPT is superior across different numbers of source domains, demonstrating its robustness to varying configurations of the source domains. Second, both GraphPrompt and GCOPE often perform worse as more datasets are added due to the negligence of structural discrepancies in various domains. In contrast, SAMGPT exhibits consistent improvement with the addition of more source domains, validating the effectiveness of our structure alignment and adaptation.

**Model ablation.** We analyze several variants of SAMGPT by removing key components, including structure tokens, holistic prompts and specific prompts. We report the results of these variants and SAMGPT in Table 4. Note that Variant 1, which lacks our structural alignment design, is equivalent to the feature alignment method MDGPT [61].

The results confirm that each component plays a critical role, as discussed below. First, the use of structure tokens is essential. Notably, Variant 3 consistently outperforms Variant 1 and 2, both of which do not employ structure tokens, demonstrating the effectiveness of structure tokens in aligning multi-domain structural knowledge. Second, removing holistic prompts leads to a drop in performance, evident from the superior accuracy of Variants 2 over Variant 1, and SAMGPT over Variant 3. This highlights the significance of incorporating holistic multi-domain topological information via holistic prompts. Third, specific prompts proves to

**Table 4: Model ablation study on key components of SAMGPT.**

| Methods | Structure tokens | Holistic prompts | Specific prompts | Target domain for node classification | | | Target domain for graph classification | | |
|---|---|---|---|---|---|---|---|---|---|
| | | | | Cora | Photo | Facebook | Cora | Photo | Facebook |
| VARIANT 1 | × | × | × | 36.36 ± 12.71 | 49.10 ± 9.94 | 35.36 ± 9.06 | 45.44 ± 13.47 | 52.45 ± 12.37 | 38.74 ± 10.26 |
| VARIANT 2 | × | × | ✓ | 40.62 ± 11.79 | 56.23 ± 9.04 | 39.80 ± 10.39 | 45.63 ± 13.52 | 57.78 ± 11.64 | 42.22 ± 10.95 |
| VARIANT 3 | ✓ | × | × | 44.26 ± 10.92 | 56.61 ± 10.14 | 41.11 ± 8.34 | 52.88 ± 12.25 | 58.14 ± 12.01 | 43.12 ± 9.76 |
| VARIANT 4 | ✓ | ✓ | × | 46.10 ± 12.02 | 57.76 ± 10.00 | 40.46 ± 8.89 | 54.52 ± 14.32 | 58.12 ± 12.30 | 43.15 ± 10.12 |
| SAMGPT | ✓ | ✓ | ✓ | 47.80 ± 11.88 | 58.71 ± 8.69 | 42.70 ± 8.73 | 55.35 ± 13.62 | **58.75** ± 11.67 | 43.71 ± 9.54 |

**Table 5: Analysis on homophilic and heterophilic graphs.**

| Target domain | Source domains | Accuracy (%) | | |
|---|---|---|---|---|
| | | GRAPHPROMPT | GCOPE | SAMGPT |
| Squi* | Cham + Corn + Cora | 18.98±4.89 | 18.98±4.75 | **20.43**±4.75 |
| Corn* | Squi + Cham + Cora | 29.67±8.36 | 27.19±8.51 | **32.57**±8.68 |
| Cham* | Squi + Corn + Cora | 23.28±4.63 | 23.24±4.50 | **23.89**±4.91 |
| Facebook | Squi + Cora + Photo | 32.22±6.91 | 35.81±7.89 | **41.10**±9.38 |

Squi*, Cham*, Corn* are short for Squirrel, Chameleon, and Cornell, respectively.

be essential, as demonstrated by Variant 4 outperforming Variant 3. This indicates the significance of source domain-specific structural knowledge for effective cross-domain adaptation. Finally, the key components, including structure tokens, holistic prompts and specific prompts are instrumental in enabling SAMGPT to achieve the best overall performance.

## 5.4 Homophily Sensitivity

Apart from feature and structural differences, graphs also exhibit varying homophily and heterophily patterns based on whether linked nodes share the same attribute [20, 59, 67]. To further assess the robustness of SAMGPT across domains with varying homophily ratios, we conduct 1-shot node classification on homophilic (*Cora*, *Photo*, *Facebook*) and heterophilic (*Chameleon*, *Cornell* and *Squirrel*) graphs[3]. We present the results in Table 5 and observe that SAMGPT consistently surpasses GRAPHPROMPT and GCOPE, regardless of whether the source or target domains are homophilic or heterophilic. These results further validate the efficacy of SAMGPT, demonstrating its ability to leverage multi-domain knowledge across a wide variety of graph domains. Note that we focus on the node classification task here, as homophily ratio is defined based on node attributes, which directly impacts node-level tasks.

## 5.5 Hyperparameter Sensitivity

We investigate the impact of hyperparameters, $\alpha$ and $\beta$, in SAMGPT. $\alpha$ governs the fusion of feature and structure alignment, as well as their adaptation, in Eqs. (7) and (12), whereas $\beta$ controls the aggregation of holistic and domain-specific adaptation in Eq. (11). We vary $\alpha$ and $\beta$ and present 1-shot node and graph classification results on three target domain, *Cora*, *Photo* and *Facebook*, in Fig. 4, with error bars denoting the standard deviation. We observe that increasing $\alpha$ from lower values initially enhances performance as structure alignment and adaptation are emphasized. However, after reaching a peak ($\alpha = 1$), accuracy begins to decline as $\alpha$ grows further,

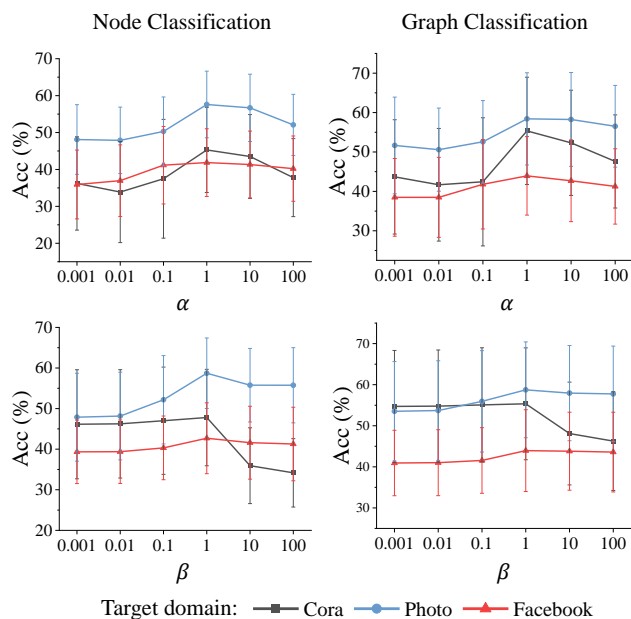

**Figure 4: Sensitivity study of $\alpha$ and $\beta$.**

implying that both feature and structure alignment are essential. Moreover, $\beta$ exhibit a trend similar to that of $\alpha$, demonstrating that incorporating both holistic and domain-specific knowledge is vital for cross-domain adaptation. Based on the above observations, we set $\alpha = 1$ in our experiments, indicating a balance between the feature and structure counterparts, and $\beta = 1$, indicating a balance between holistic and specific prompts, both of which show robust empirical performance.

## 6 Conclusions

In this paper, we propose SAMGPT, a graph foundation model with structure alignment for text-free multi-domain graph pre-training and cross-domain adaptation. In the pre-training phase, SAMGPT utilizes a series of structure tokens to harmonize the structural distributions across multiple source domains and to extract multi-domain topological information. For downstream cross-domain adaptation, SAMGPT employs dual prompts, holistic prompts and specific prompts, to tailor pre-trained holistic and domain-specific topological knowledge, respectively, to the target domain. We conduct extensive experiments on seven public benchmark datasets, demonstrating that SAMGPT significantly outperforms a variety of state-of-the-art baseline methods.

---

[3]We present details about heterophilic datasets in Appendix F.

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

# Appendices

## A    Algorithm

In multi-domain pre-training phase, we first apply dimension alignment method $DAL$ to align feature dimensions from different source domains by Eq. (3). Then, we use a feature alignment method to unify feature semantic spaces by Eq. (4). For structure alignment, we inject source domain-specific structure tokens into each layer of the graph encoder by Eq. (6)n. Finally, we fuse the feature aligned embedding and structure aligned embedding by Eq. (7), and optimize pre-training loss by Eq. (8).

We further present the key steps for cross-domain adaptation in Algorithm 1. In lines 3–4, we align target domain feature dimensions with source domains. In lines 6–7, we integrate feature adaptation method to generate feature-level adapted embeddings. In lines 8–22, we employ dual prompts to adapt structural prior knowledge to target domain. Specifically, we first inject holistic prompts to modify the structure-based aggregation in each layer of the graph encoder for holistic knowledge adaptation (lines 9–13). Then, we generate specific prompts by fusing the pre-trained structure tokens (lines 14–16), and utilize specific prompts for domain-specific knowledge adaptation (lines 18–20). We obtain structure-level adapted embeddings by fusing holistic and domain-specific embeddings (lines 21–22), and generate final embeddings by aggregating feature- and structure-level adapted embeddings (lines 23–24). Finally, we update the embeddings for the prototypical instances based on the labeled samples in the task (lines 25–27) and optimize holistic prompts, $\Lambda$ and $\Gamma$ (lines 28-29). Note that updating prototypical is required exclusively for classification tasks.

## B    Complexity Analysis

For a downstream graph $G_T = (V_T, E_T, \mathbf{X}_T)$ from the target domain $D_T$, the computational process of structure adaptation are constructed by injecting holistic prompts and specific prompts to the process of encoding nodes via a pre-trained GNN. In a standard GNN, each node aggregates messages from up to $n$ neighbors per layer. Assuming the aggregation involves at most $n$ neighbors, the complexity of calculating node embeddings over $L$ layers per batch time is $O(n^L \cdot |V_T|)$. holistic prompts are directly injected to each layer of the GNN, leading to a complexity of $O(L \cdot |V_T|)$. specific prompts are first generated by pre-trained structure tokens, with a complexity of $0(L \cdot K)$. Then specific prompts modify the structure-base aggregation with a complexity of $O(L \cdot |V_T|)$. Since holistic prompts and specific prompts modifies the node encoding phase separately, the overall complexity is $(2(L + n^L) \cdot |V_T| + L \cdot K)$. Thus, the encoding phase by pre-trained GNN dominates the overall complexity, as $O(2n^L \cdot |V_T|)$ far exceeds $O(2L \cdot (|V_T| + K))$.

## C    Further Descriptions of Datasets

In this section, we provide a summary of these datasets in Table 6 and a further comprehensive descriptions of these datasets. Specifically, average node degree, average shortest path length [3], and average clustering coefficient [6] reflect the structural properties of various datasets/domains, from which we observe that different domains exhibit unique structural characteristics, highlighting

---

**Algorithm 1** CROSS-DOMAIN ADAPTATION FOR SAMGPT

**Input:** Pre-trained graph encoder $GE$ with parameters $\Theta_{\text{pre}}$, pre-trained structure tokens $\mathcal{T}_{\text{pre}}$, target domain dimension alignment method $\text{DA}_T(\cdot)$ and feature adaptation methods $\text{FAD}(\cdot)$
**Output:** Optimized holistic prompts $\mathcal{P}_{\text{hol}}$, coefficient $\Lambda$, and $\Gamma$

1: **while** not converged **do**
2:     **for** each graph $G_T = (V_T, E_T, \mathbf{X}_T)$ in target domain $D_T$ **do**
3:         /* Target domain feature dimensions alignment by Eq. (3) */
4:         $\tilde{\mathbf{X}} \leftarrow \text{DAL}_T(\mathbf{X})$
5:         $\mathcal{P}_{\text{hol}}, \Lambda, \Gamma \leftarrow$ initialization
6:         /* Feature adaptation by Eq. (5) */
7:         $\mathbf{H}^{\text{FAD}} \leftarrow \text{GE}(\text{FAD}(\mathcal{G}, \tilde{\mathbf{X}}; \Gamma); \Theta_{\text{pre}})$
8:         /* Structure alignment by dual prompts */
9:         /* Adaptation of holistic structural prior knowledge */
10:        **for** each layer in $GE$ **do**
11:           /* Modification to $GE$ via holistic prompts by Eq. (9) */
12:           $\mathbf{h}_v^l \leftarrow \text{Aggr}(\mathbf{h}_v^{l-1}, \{\mathbf{p}_{\text{hol}}^l \odot \mathbf{h}_u^{l-1} : u \in \mathcal{N}_v\}; \theta_{\text{pre}}^l), \ \forall v \in G_T$
13:        $\mathbf{H}^{\text{hol}} \leftarrow \text{STACK}(\mathbf{h}_v : \forall v \in G_T)$
14:        /* Generation of specific prompts by Eq. (10) */
15:        **for** $\mathbf{p}_{\text{spe}}^l$ in $\mathcal{P}_{\text{spe}}$ **do**
16:           $\mathbf{p}_{\text{spe}}^l \leftarrow \sum_{i=1}^K \lambda_i^l \mathbf{t}_{S_i}^l$
17:        /* Modification to $GE$ via specific prompts*/
18:        **for** Each layer in $GE$ **do**
19:           $\tilde{\mathbf{h}}_v^l \leftarrow \text{Aggr}(\tilde{\mathbf{h}}_v^{l-1}, \{\mathbf{p}_{\text{spe}}^l \odot \tilde{\mathbf{h}}_u^{l-1} : u \in \mathcal{N}_v\}; \theta_{\text{pre}}^l), \ \forall v \in G_T$
20:        $\mathbf{H}^{\text{spe}} \leftarrow \text{STACK}(\tilde{\mathbf{h}}_v : \forall v \in G_T)$
21:        /* Fusion of dual prompts tuned embeddings by Eq. (11) */
22:        $\mathbf{H}^{\text{SAD}} \leftarrow \mathbf{H}^{\text{hol}} + \beta \mathbf{H}^{\text{spe}}$
23:        /* Fusion of adapted embeddings by Eq. (12) */
24:        $\mathbf{H}^{\text{AD}} \leftarrow \mathbf{H}^{\text{FAD}} + \alpha \mathbf{H}^{\text{SAD}}$
25:        /* Update prototypical nodes */
26:        **for** each class $y$ **do**
27:           $\mathbf{h}_y \leftarrow \text{AVERAGE}(\mathbf{h}_x : \text{instance } x \text{ belongs to class } y)$
28:        /* Optimizing $\mathcal{P}_{\text{hol}}$, $\Lambda$, and $\Gamma$ */
29:        Calculate $\mathcal{L}_{\text{down}}(\Omega; \mathcal{P}_{\text{hol}}, \Lambda, \Gamma)$ by Eq. (13)
30: **return** $\mathcal{P}_{\text{hol}}$, $\Lambda$, and $\Gamma$

---

the importance of structural alignment in both multi-domain pre-training and cross-domain adaptation.

- *Cora* [22] consists of 2,708 publications in the computing field, each categorized into one of seven classes. The citation network comprises 5,429 edges. Each publication is represented by a binary word vector indicating the presence or absence of words from a dictionary containing 1,433 unique words.
- *Citeseer*[32] contains 3,312 computer science publications, each belonging to one of six categories, distinct from those in *Cora*. The citation network consists of 4,732 edges. Each publication is represented by a binary word vector, reflecting the presence or absence of words from a dictionary of 3,703 unique words.
- *PubMed* [32] consists of 19,717 biomedical publications related to diabetes, each classified into one of three categories. The citation network includes 44,338 edges. Each publication is represented by a TF/IDF-weighted word vector, indicating the presence of 500 unique words from the dictionary.

**Table 6: Summary of datasets.**

| | Nodes | Edges | Feature dimension | Node classes | Avg nd | Avg spl | Avg cc |
|---|---|---|---|---|---|---|---|
| Cora | 2,708 | 10,556 | 1,433 | 7 | 3.89 | 6.30 | 0.24 |
| Citeseer | 3,327 | 9,104 | 3,703 | 6 | 2.73 | 9.31 | 0.14 |
| Pubmed | 19,717 | 88,648 | 500 | 3 | 4.49 | 6.33 | 0.06 |
| Photo | 7,650 | 238,162 | 745 | 8 | 31.13 | 4.05 | 0.40 |
| Computers | 13,752 | 491,722 | 767 | 10 | 35.75 | 3.38 | 0.34 |
| Facebook | 22,470 | 342,004 | 128 | 4 | 15.22 | 4.97 | 0.35 |
| LastFM | 7,624 | 55,612 | 128 | 18 | 7.29 | 5.23 | 0.21 |

Avg is short for average. nd stands for node degree. spl denotes shortest path length. cc represents clustering coefficient.

- *Photo* [33] contains 7,487 products related to photography, each assigned to one of eight categories. The co-purchase network comprises 119,043 edges, representing products frequently bought together. Each product is described by a feature vector derived from its metadata and reviews, and is labeled according to its category.
- *Computers* [33] includes 13,752 computer-related products, divided into ten categories. The co-purchase network consists of 245,861 edges, representing products that are frequently bought together. Each product is characterized by a feature vector generated from its metadata and reviews and is labeled according to its respective category.
- *Facebook* [30] represents a page-to-page graph of verified Facebook sites. The nodes correspond to official Facebook pages, and the edges indicate mutual "likes" between these pages. Node features are derived from the descriptions provided by the page owners to outline the purpose of their sites.
- *LastFM* [31] represents a social network of LastFM users, collected via the public API in March 2020. The nodes correspond to LastFM users from various Asian countries, and the edges represent mutual follower relationships. The node features are extracted based on the artists that users have liked. The associated task for this graph is multinomial node classification, where the objective is to predict each user's location, derived from the country field in their profile.

## D Further Descriptions of Baselines

In this section, we provide additional details about the baselines used in our experiments.

(1) **End-to-end GNNs**

- **GCN** [12]: GCN employs a mean-pooling approach for neighborhood aggregation, enabling the integration of information from adjacent nodes.
- **GAT** [40]: GAT relies on neighborhood aggregation for node representation learning, but distinguishes itself by assigning varying attention weights to neighbors, thus adjusting their influence on the aggregation process.

(2) **Graph Pre-training Models**

- **DGI** [40]: DGI is a self-supervised pre-training approach. It is based maximizing mutual information (MI), with the goal of strengthening the MI between local node representations and their global context.
- **InfoGraph** [35]: Building on DGI, InfoGraph focuses on graph-level tasks, aiming to align node and graph embeddings by maximizing the similarity between them.
- **GraphCL** [55]: GraphCL applies various graph augmentations for self-supervised learning, leveraging structural patterns within graphs. Its main objective is to improve the similarity across different augmentations during pre-training.
- **GPPT** [36]: GPPT pre-trains a GNN model via link prediction task. Its downstream prompt module is specifically designed for node classification, unifying it with the pre-training link prediction task.
- **GPF** [5]: GPF serves as a universal prompt-based tuning approach for pre-trained graph models. It adapts the input graph's feature space to simulate the behavior of various prompting functions.
- **GraphPrompt** [17]: GraphPrompt utilizes subgraph similarity calculations as a unified framework to bridge the gap between pre-training and downstream tasks, supporting both node and graph classification. During downstream adaptation, a learnable prompt is tuned to incorporate task-specific knowledge.

(3) **Graph Cross-domain Models**

- Hassani [8]: Hassani proposes an attention-based graph encoder that leverages both contextual and topological views to capture task-specific information for quick adaptation, as well as task-independent knowledge for efficient transfer across domains.

(4) **Multi-Domain Graph Pre-Training Model**

- **GCOPE** [65]: GCOPE propose a multi-domain pre-training strategy that integrates graph datasets from various domains using domain-specific interconnecting virtual nodes, which link nodes within the same domain. The main objective is to enhance downstream performance by harnessing knowledge from multiple source domains.

## E Implementation Details

**General settings Optimizer:** For all experiments, we use the Adam optimizer.

**Environment:** The environment in which we run experiments is:

- Operating system: Ubuntu 22.04.2
- CPU information: AMD EPYC 7742 64-Core Processor
- GPU information: NVIDIA GeForce RTX 3090 (24 GB)

**Details of baselines.** We utilize the officially provided code for all open-source baselines. Each model is tuned based on the settings recommended in their respective literature to achieve optimal performance.

For the baseline GCN [12], we employ a 3-layer architecture, and set the hidden dimensions to 256. For GAT [40], we employ a 2-layer architecture and set the hidden dimension to 64. Additionally, we apply 8 attention heads in the first GAT layer.

For DGI [40], we utilize a 1-layer GCN as the backbone and set the hidden dimensions to 256. Additionally, we employ prelu as the activation function. For InfoGraph [35], a 3-layer GCN is used as the backbone, with its hidden dimensions set to 256. For GraphCL [55], a 1-layer GCN is also employed as its backbone, with the hidden dimensions set to 256. Specifically, we select edge dropping as the augmentations, with a default augmentation ratio of 0.2. For GPPT [36], we utilize a 2-layer GraphSAGE as its backbone, setting the hidden dimensions to 256. We employ a mean aggregator for the aggregation in the backbone. For GraphPrompt [17], a 3-layer GCN is used as the backbone for all datasets, with the hidden dimensions set to 256. For GPF [5], employs a 5-layer GCN as the backbone for all datasets, following the recommended settings. The hidden dimensions are set to 256.

For Hassani [8],We employ a 3-layer GCN is used as the backbone for all datasets, with the hidden dimensions set to 256.

For GCOPE [65], we employ a 2-layer GCN as the backbone and set the hidden dimensions to 100. Downstream adaptation is achieved through fine-tuning, as it is reported to yield the best performance in their literature.

For all baselines except for GCOPE, we set the unified feature dimensions to 50, the same as our SAMGPT. For GCOPE, we adhere to the recommended settings and set the unified feature dimensions to 100.

**Details of SAMGPT.** For our proposed SAMGPT, we utilize a 3-layer GCN as the backbone for all datasets, with the hidden dimensions set to 256. We set the unified feature dimensions to 50.

## F   Details about Heterophilic Datasets

To evaluate the robustness of SAMGPT across graphs with varying homophily ratios, we conduct experiments on both homophilic and heterophilic datasets as in Sect. 5.4. Details of the heterophilic datasets are introduced as follows. (1) *Chameleon* [30] is a Wikipedia-based network containing 2,277 pages, categorized into five groups based on their average monthly traffic. This dataset forms a network with 36,101 edges, and the node features are derived from key nouns extracted from the Wikipedia content. The homophily ratio is 0.23. (2) *Cornell* [25] is another webpage network consisting of 183 nodes, where each node represents a webpage, and 295 edges denoting hyperlinks between them. The node features are derived from a bag-of-words representation of the webpages. These pages are manually classified into five categories: student, project, course, staff, and faculty. The homophily ratio is 0.22. (3) *Squirrel* [30] consists of 5,201 Wikipedia pages discussing specific topics. The pages are divided into five categories based on their average monthly traffic. This dataset forms a page-to-page network with 217,073 edges, and the node features are derived from various informative nouns present in the Wikipedia content. The homophily ratio is 0.30.

## G   Data Ethics Statement

To evaluate the efficacy of SAMGPT, we conducted experiments which only use publicly available datasets, namely, Cora[4], Citeseer[5], Pubmed[6], Photo[7], and Computers[8], Facebook[9], LastFM[10], Chameleon[11], Cornell[12] and Squirrel[13] in accordance to their usage terms and conditions if any. We also confirm that no personally identifiable information was utilized, and this research did not involve any human or animal subjects.

---

[4]https://github.com/shchur/gnn-benchmark/raw/master/data/npz/cora.npz
[5]https://github.com/shchur/gnn-benchmark/raw/master/data/npz/citeseer.npz
[6]https://github.com/shchur/gnn-benchmark/raw/master/data/npz/pubmed.npz
[7]https://github.com/shchur/gnn-benchmark/raw/master/data/npz/amazon_electronics_photo.npz
[8]https://github.com/shchur/gnn-benchmark/raw/master/data/npz/amazon_electronics_computers.npz
[9]https://graphmining.ai/datasets/ptg/facebook.npz
[10]https://graphmining.ai/datasets/ptg/lastfm_asia.npz
[11]https://github.com/SitaoLuan/ACM-GNN/tree/main/new_data/chameleon
[12]https://github.com/bingzhewei/geom-gcn/tree/master/new_data/cornell
[13]https://github.com/SitaoLuan/ACM-GNN/tree/main/new_data/squirrel

