# OpenReview forum: "SAMGPT: Text-free Graph Foundation Model for Multi-domain Pre-training and Cross-domain Adaptation"
_ACM.org/TheWebConf/2025/Conference — WWW 2025 Oral_

### Official Review · Reviewer_cCq8 · 2024-11-20

**Novelty:** 5
**Technical Quality:** 5

**Review:**

### Summary:

The paper presents a novel text-free graph foundation model designed for multi-domain pretraining and cross-domain adaptation. Comprehensive experiments demonstrate its effectiveness across various datasets, though some areas could benefit from further exploration and refinement.

### Pros:

1. SAMGPT addresses a critical gap in graph foundation models by focusing on text-free graphs.
2. The dual prompt strategy effectively combines multi-domain structural knowledge with domain-specific adaptations, enhancing cross-domain learning.
3. The experiments are thorough and demonstrate the effectiveness of the proposed method.

### Cons:

1. SAMGPT's dependence on aligning features and structures raises concerns about its suitability for domains with significant structural differences.
2. There is no experimental validation to determine the most suitable unified dimension in the module of DAL(why choose 50?).
3. The paper does not evaluate with different backbones (e.g. GraphSAGE or GIN) for a more adaptability view on SAMGPT.
4. Some minor mistakes should be corrected, for example, in lines 79,1203 and 1206.

**Questions:**

Refer to Cons.

**Reviewer Confidence:**

3: The reviewer is confident but not certain that the evaluation is correct

**Scope:**

4: The work is relevant to the Web and to the track, and is of broad interest to the community

---

### Official Review · Reviewer_VqC8 · 2024-11-28

**Novelty:** 5
**Technical Quality:** 5

**Review:**

This paper introduces SAMGPT, a novel approach for acquiring knowledge from multiple source domains and adapting models to unseen domains in downstream tasks. By leveraging the graph prompting paradigm, SAMGPT incorporates innovative techniques, including structure tokens and dual prompts, to enhance domain adaptation.

The proposed method offers a simple yet effective solution for multi-domain adaptation. It utilizes a dual-prompt strategy that separates holistic and domain-specific prompts, facilitating better adaptation across diverse domains. The pipeline is built upon three sets of learnable parameters: (1) structural alignment parameters during pre-training, (2) holistic parameters during prompting, and (3) domain-specific parameters during prompting. While the third set of parameters follows common practices in graph prompting, the first two represent novel contributions that bridge the gap between pre-training and fine-tuning. Extensive experiments demonstrate the effectiveness of the SAMGPT framework, showing significant performance improvements over existing baselines.

Drawbacks:
- Real-Time Efficiency and Space Complexity Analysis. The paper does not provide a detailed real-time efficiency analysis of SAMGPT in comparison to other baselines. Including an evaluation of runtime performance, particularly under varying domain adaptation scenarios, would provide deeper insights into the practicality of SAMGPT for real-world applications. Additionally, a comprehensive analysis of space complexity, as well as the total number of tunable parameters, would strengthen the evaluation by giving a clearer picture of the computational cost and scalability of the proposed approach.
- Omission of Key Related Work. A significant related work, Inductive Graph Alignment Prompt: Bridging the Gap between Graph Pre-training and Inductive Fine-tuning from a Spectral Perspective (WWW 2024), is not discussed in the current manuscript. This work focuses on inductive learning in graph prompting methods and offers valuable insights that would complement the discussion on SAMGPT. Including a reference to this study would enhance the completeness of the related work section and provide a more comprehensive context for the proposed approach.

**Questions:**

The authors assert in line 112 that existing works overlook structural differences across domains. They further argue in lines 96–99 that graph structures from different datasets are unique and may not be directly transferable. These claims, however, seem somewhat unclear. Is there a more in-depth analysis or explanation provided to substantiate these assertions?

**Reviewer Confidence:**

4: The reviewer is certain that the evaluation is correct and very familiar with the relevant literature

**Scope:**

4: The work is relevant to the Web and to the track, and is of broad interest to the community

---

### Official Review · Reviewer_KYij · 2024-12-02

**Novelty:** 4
**Technical Quality:** 5

**Review:**

The paper tackles multi-domain graph neural network pre-training and domain adaptation. The authors proposed using learnable structure tokens to align domain-specific patterns and a dual prompt mechanism to adapt pre-trained knowledge. While the method builds heavily on existing prompt-tuning and structure-aware graph learning concepts, it provides a working solution with consistent empirical gains.

Strengths:
1. Well-structured technical approach combining structure alignment and prompt learning in a principled way
2. Abundant experiments and ablation studies showing consistent improvements across datasets and effect of model components
3. The authors offered their code and data for reproducibility
4. The paper is well written and easy to follow

Weaknesses:
1. Even though technical innovations are made for the multi-domain setting, they are arguably quite limited. The structure tokens are essentially learnable masks applied to aggregation, and arguably similar to attention mechanisms and virtual nodes in recent works
2. The model does not appear to scale very well. The model apparently uses full batch training. In the authors' implementation, all graphs from all training domains has to be loaded into the GPU memory. The experiments were all conducted on relatively small scale datasets. Moreover, the model does not scale well with the number of domains because each domain introduces additional model parameters and increased hidden dimensions. These efficiency issues significantly undermines the "foundational" aspect of the proposed method.
3. The proposed method is only tested on node and graph classification.

Overall, despite the clear limitations, I think the paper holds its value in multi-domain GNN pre-training and I lean toward acceptance. The work will benefit the community with its empirical validation, clear presentation, and reproducibility.

**Questions:**

Can you address the scalability concerns?

**Reviewer Confidence:**

3: The reviewer is confident but not certain that the evaluation is correct

**Scope:**

4: The work is relevant to the Web and to the track, and is of broad interest to the community

---

### Official Review · Reviewer_Un6B · 2024-12-02

**Novelty:** 5
**Technical Quality:** 5

**Review:**

The paper demonstrates a solid understanding of the challenges in multi-domain graph pre-training and cross-domain adaptation, particularly for text-free graphs.They propose a novel Structure Alignment framework for text-free Multi-domain Graph Pre-Training and cross-domain adaptation (SAMGPT). It is designed to learn multi-domain knowledge from graphs originating in multiple source domains, which can then be adapted to address applications in an unseen target domain. they propose the notion of structure tokens to align structural distributions across multiple domains.  Each domain is equipped with a series of structure tokens, which modify the structure-based aggregation in each layer of the graph encoder. The proposed SAMGPT framework is well-designed and addresses key limitations in existing methods, such as structural differences across domains. Comprehensive experiments on seven public datasets provide a robust evaluation of the framework's effectiveness.The presentation of the framework is generally clear, with logical structuring and well-defined components. This work is significant as it tackles a challenging problem with practical implications for numerous real-world applications, such as recommendation systems, social networks, and knowledge graphs. The ability to adapt to unseen domains without relying on textual attributes expands the applicability of graph-based models.

**Questions:**

(1) How well do structure tokens handle graphs with highly irregular or noisy structures? Are there scenarios where this approach may fail or require additional adjustments?
﻿
(2) How does the number of source domains affect the scalability of specific prompts? Could an increasing number of domains lead to overfitting or computational challenges?

**Reviewer Confidence:**

4: The reviewer is certain that the evaluation is correct and very familiar with the relevant literature

**Scope:**

3: The work is somewhat relevant to the Web and to the track, and is of narrow interest to a sub-community

---

### Official Review · Reviewer_EWdj · 2024-12-04

**Novelty:** 4
**Technical Quality:** 3

**Review:**

The paper presents SAMGPT, a graph foundational model designed for multi-domain pre-training and cross-domain adaptation of text-free graphs. It introduces structure tokens and dual prompts to harmonize structural variations across domains during the pre-training phase and to facilitate domain-specific adaptations during downstream tasks. The model's effectiveness is demonstrated through extensive experiments on seven public datasets,

Strengths:

1. The paper is well-written and easy to follow.
2. Extensive experiments across seven public datasets demonstrate the model's superior performance compared to state-of-the-art methods.
3. The provision of code and data links supports reproducibility

Weaknesses:

1. The idea is easy-to-get, seems lack novelty to me. The technique contribution is limited.
2. Why is text-free a notable tag to the proposed methods? aren’t all most graph neural network related researches, including those pre-training and fine-tuning works, conducted on text-free data?
3. There is no specific optimization objective proposed to specifically address the cross-domain discrepancy problem, which is a critical issue for multi-domain pre-training. For example, those Out-of-distribution generalization techniques can be used.

**Questions:**

1. Please introduce the feature alignment and feature adaptation in Eq.4 and 5 in detail. The reference Yu et al.[56] appear to be a survey, and I could not find a specific definition of them in this paper.

2. Why is the prompt named structure tokens, while it is conducted on feature embeddings?

**Reviewer Confidence:**

3: The reviewer is confident but not certain that the evaluation is correct

**Scope:**

4: The work is relevant to the Web and to the track, and is of broad interest to the community